# Restoration of Individual Tree Missing Point Cloud Based on Local Features of Point Cloud

## Wei Cao, Jiayi Wu, Yufeng Shi * and Dong Chen

Department of Geomatics, Nanjing Forestry University, Nanjing 210037, China; caowei@njfu.edu.cn (W.C.); jiayiwu@njfu.edu.cn (J.W.); chendong@njfu.edu.cn (D.C.)
* Correspondence: yfshi@njfu.edu.cn

**Abstract:** LiDAR (Light Detection And Ranging) technology is an important means to obtain three-dimensional information of trees and vegetation. However, due to the influence of scanning mode, environmental occlusion and mutual occlusion between tree canopies and other factors, a tree point cloud often has different degrees of data loss, which affects the high-precision quantitative extraction of vegetation parameters. Aiming at the problem of a tree laser point cloud being missing, an individual tree incomplete point cloud restoration method based on local features of the point cloud is proposed. The $L_1$-Median algorithm is used to extract key points of the tree skeleton, then the dominant direction of skeleton key points and local point cloud density are calculated, and the point cloud near the missing area is moved based on these features to gradually complete the incomplete point cloud compensation. The experimental results show that the above repair method can effectively repair the incomplete point cloud with good robustness and can adapt to the individual tree point cloud with different geometric structures and correct the branch topological connection errors.

**Keywords:** LiDAR; individual tree; incomplete point cloud; tree skeleton; local feature; $L_1$-Median algorithm

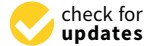



## 1. Introduction

The 3D information of vegetation is an important part of digital forestry, which provides important data guarantees for vegetation carbon storage, biomass and ecological assessment. Under the background of rapid development of big data and artificial intelligence, active remote sensing technology represented by LiDAR has gradually become an important means for vegetation 3D information acquisition and forest resources investigation. It provides technical support for forest resource investigation and ecological process detection and analysis at different scales [1–6]. LiDAR technology is characterized by the fast acquisition of spatial data, a high degree of automation, high precision and a large amount of data [7]. With the help of this technology, high-precision spatial information of individual trees and small-scale forests can be quickly obtained, and then the segmentation of individual tree canopy structures [8–10] and tree structural reconstructions [4,11], and the extraction of individual tree vegetation information [12,13] can be performed. For example, Terrestrial Laser Scanning (TLS) can quickly extract vegetation parameters, such as tree volume, leaf area index and gap fraction, providing key parameters for forest ecological evaluation investigation [14]. In addition, TLS can also be used to evaluate the height of individual trees and leaf area density [15]. In addition to individual tree/forest vegetation parameter extraction, high-precision and high-density tree point cloud is also widely used in high-precision extraction and reconstruction of branch structure [16], which can provide important data and model support for digital forestry and digital city construction.

However, restricted by factors, such as laser scanning mode and environmental occlusion, the LiDAR point cloud of individual trees/forests is often missing, to varying degrees. For example, (1) limited by the scanning resolution, the complex layered branch structure

inside the canopy often has the situation that the thin branches in the center of the canopy are blocked by the thin branches in the outer layer; (2) affected by the scanning point cloud density and scanning distance, different trees/different parts of the same tree due to its distance inconsistency, often result in a point cloud density that is not uniform, which to a certain extent, can be seen as an incomplete point cloud. In addition, the point cloud near the missing area also has problems, such as drastic density changes; (3) the thin branch point cloud inside the canopy is usually missing in a large range due to the self-occlusion of branches/leaves and the external occlusion of pedestrians/billboards during the scanning process; (4) compared with the time-consuming and labor-intensive multi-station scanning, single-station scanning is simple and efficient, but the data loss of a point cloud obtained by single-station scanning is more serious. Usually, only the tree point cloud facing the scanning device can be obtained, and a large number of missing point clouds exist on the side away from the scanner.

The absence of point clouds usually has an adverse impact on the extraction of vegetation parameters and the abstract expression of branches. For example, the individual tree point cloud based on airborne LiDAR, due to its lack of twigs and trunks inside the tree canopy, often produces large data deviations in large-scale ecological studies of forest biomass and volume, which brings difficulties to practical analysis and application. In addition, in the practical application of individual tree 3D reconstruction, the lack of regional point clouds will further aggravate the difficulty of high-precision 3D reconstruction of individual trees. If the incomplete point cloud is ignored and the model is directly constructed, problems, such as the wrong connection of branch topology and inconsistency of branch radius with the true value will often occur. In order to ensure the realism and fidelity of the model, point cloud repair for the missing area is one of the simplest and most direct strategies to solving the problem of high-precision model reconstruction in the case of missing parts of point clouds. Figure 1 shows the scenarios that may occur when there are missing data in the bifurcated branch structure region of an individual tree. It can be seen that when the branch point cloud is complete, the branch skeleton can be extracted correctly and completely, and the branch topological connection is correct, as shown in Figure 1a,c. However, when the point cloud inside the branch, especially the point cloud at the junction, is missing, the extracted skeleton may have topological connection errors, as shown in Figure 1b,d. In this case, if the skeleton/tree 3D model is constructed directly without data repair, problems, such as branch topology errors and radius calculation errors will often occur, affecting the accuracy of tree skeleton extraction and model reconstruction. Therefore, it is important to construct a branch point cloud enhancement algorithm based on the acquired tree point clouds to maintain the tree skeleton topology and tree model realism.

To solve the problem of missing parts of a tree point cloud, many authors have proposed data enhancement and recovery methods for tree point clouds, which can be roughly divided into the following three categories: (1) Repairing missing parts of point clouds based on point cloud features, usually using point cloud local features, such as normal vector [17], curvature factor [18], reflection intensity [19], point cloud density and normal information [20]. For example, repairing the incomplete point cloud based on structure-aware global optimization algorithms [21], based on point cloud local density information [22], and based on point cloud normal information [23]. The branch point cloud repaired by this method is often more consistent with the natural growth pattern of vegetation, but the time cost is high. (2) Multi-source data fusion for missing parts of point cloud data enhancement, such as improving point cloud data based on backpack LiDAR scanning [24] or multi echo-recording mobile laser scanning [25]. Although these methods are direct and convenient, it brings problems, such as multi-source data fusion and registration. (3) Based on prior knowledge or modeling algorithms, such as methods based on first reconstructing the branch model/fitting the branch cylinder and then completing the missing point cloud [7,26]. These methods can obtain relatively complete branch detail information, but the branch radius will violate the natural growth rules, and the subsequent

completion needs further prior constraints to ensure the realism of the branch point cloud reconstruction model.

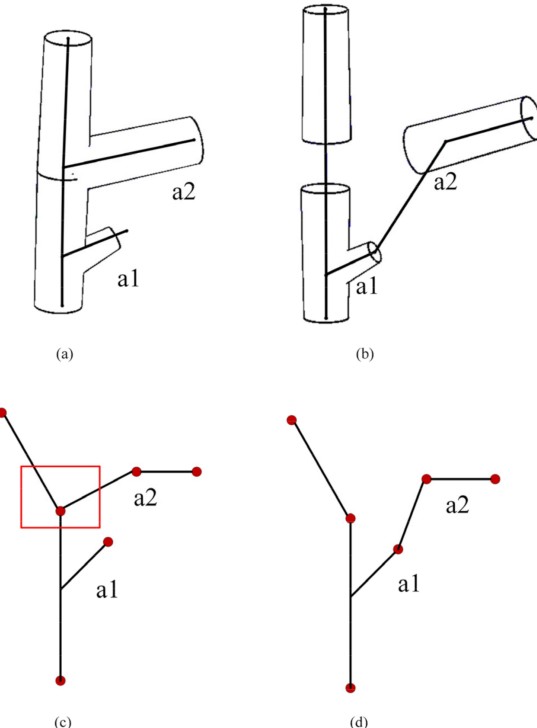

**Figure 1.** Schematic diagram of topological connection of bifurcation structure of the branch. (**a**) True trunk skeleton; (**b**) Structural fracture of branches; (**c**) True trunk skeleton line; (**d**) Missing branches causes skeleton errors. In the figure, black lines are skeleton lines, a1 and a2 are branches, red dots in (**c**,**d**) are skeleton points, and the red frame is the region where parts of the point cloud are missing.

Therefore, in order to ensure the reliability of vegetation parameter extraction and ecological analysis and the accuracy of branch reconstruction, it is important to explore a high-precision point cloud restoration algorithm that conforms to the tree structure rules. Considering the disadvantages of long recovery time of algorithms based on point cloud density and normal information, and the disadvantages of the lack of realism based on an a priori strategy, this paper adds constraints on tree structure direction and proposes an iterative point cloud optimization algorithm based on local point cloud weight density and skeleton point dominant direction. The algorithm realizes the repair and enhancement of the incomplete point cloud of an individual tree through iteration and obtains the point cloud of an individual tree branch that matches the geometry of the real tree. The repaired and enhanced tree point cloud can lay an important data foundation for the subsequent extraction of vegetation parameters and ecological analysis, as well as the reconstruction of branch structure with high accuracy and correct topological connection.

## 2. Tree Point Cloud Restoration

The process framework of the repair algorithm in this paper is shown in Figure 2. Firstly, the $L_1$-Median algorithm [27] is used to extract the median points of the individual tree point cloud as the initial skeleton key points. Then, the initial skeleton key points are combined with the original individual tree point cloud, which is used to calculate the local weight density of the incomplete point cloud and the dominant direction of each skeleton key point. They are then used as a reference for point cloud repair optimization. The repair process draws on the point cloud movement strategy of the local awareness global optimization algorithm; the force and contraction constraints of each point are added to make the adjacent points in the missing area move to the missing branches along the

constraint distance. Since there is no heterogenous point cloud involved in the restoration process, the total amount of point cloud of repaired branches remains unchanged, so the point cloud density will gradually decrease with the extension of branches, and the density change is closer to the natural rules of vegetation growth. Finally, we iterate the process of "skeleton point extraction—point cloud superposition—local point cloud feature calculation—input point cloud spatial position optimization". When the force and contractive constraint of each point near the missing area reach equilibrium, the iterative process stops and the missing repair is completed.

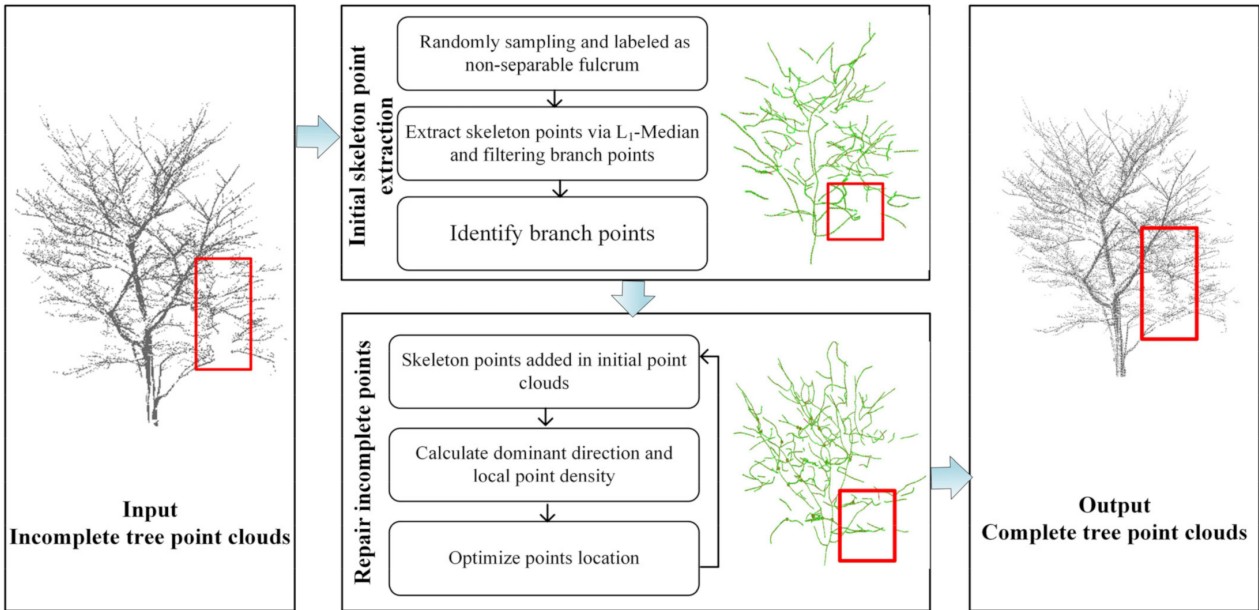

**Figure 2.** The process of incomplete point cloud repair optimization.

### 2.1. Extraction of Initial Skeleton Key Points

Studies on tree skeleton extraction based on LiDAR point cloud can be roughly divided into the following three categories: (1) Skeleton extraction based on clustering algorithms, such as the clustering algorithm based on horizontal data sets [28,29], i.e., making horizontal slices at a certain distance in the Z dimension, followed by clustering the point clouds within the horizontal slices to form skeleton points. There are also extraction methods based on K-means clustering of tree skeleton points [30,31]. (2) Skeleton extraction based on graph theory methods, such as tree skeleton extraction based on octree structure [32,33]. (3) Laplacian operator-based skeleton extraction methods [5,34].

#### 2.1.1. Random Sampling of Original Point Cloud

Considering that the skeleton extraction involved in this paper only serves to calculate the local features and repair of a point cloud, and does not require high spatial geometric accuracy, the $L_1$-Median algorithm proposed in reference [27] is selected as the skeleton key point extraction method. Since there is often a discrepancy in cloud density between twigs and main branches, direct skeleton extraction based on the $L_1$-Median algorithm is prone to shrinkage inconsistency, that is, skeleton extraction is completed with a large number of skeleton points with higher density, while skeleton points of twig are not effectively extracted. In order to prevent such a situation, this paper refers to the iterative shrinkage method in reference [27], which firstly identifies the labeled skeleton branch points, then selects suitable bridging points from the non-branch points to connect the branch points with the non-branch points, and finally gradually expands the neighborhood values to achieve the growth and merging of branches. Specifically, the original individual tree point

cloud is first randomly sampled to obtain a set of sampled points, and the set of points is marked as non-branching points.

### 2.1.2. Point Cloud Skeleton Extraction

After random sampling of the original point set, the tree skeleton key points are extracted based on the $L_1$-Median algorithm with iterative shrinkage. Specifically, a scattered individual tree point cloud that is undirected, unevenly distributed, and contains noise and outlier points is used as input, denoted as $Q = \{q_j\}_{j \in J}$, and the output of the algorithm is a one-dimensional curve skeleton point. In order to extract the initial skeleton point of an individual tree point cloud, this paper transforms the initial skeleton point location problem into a location problem of finding a set of optimal point sets $X = \{x_i\}_{i \in I}$, where point set $X$ is a set of point with the minimum Euclidean distance from the point cloud in its local neighborhood. The formula for a point $x_i$ in the point set $X$ is as in Equation (1) [22,27]:

$$x_i = \underset{X}{\mathrm{argmin}} \sum_{i \in I} \sum_{j \in J} \parallel x_i - q_j \parallel \theta(\parallel x_i - q_j \parallel) + R(X) \,, \tag{1}$$

where the first term is to calculate the spatial position of the optimal point set $X$ in the input individual tree point set $Q$; the second term $R(X)$ is a regular term with conditions attached, which mainly serves to generate a repulsive force when the local branch skeleton is formed and imposes a penalty on the position of point $x_i$ to ensure the uniform distribution of the skeleton point positions. $I$ is the index point set of point set $X$ and $J$ is the index point set of point set $Q$. $\theta$ is a fast decaying function, whose definition is shown in Equation (2):

$$\theta = e^{-r^2/(h/2)^2} \,, \tag{2}$$

where $h$ is the local support radius and it defines the size of the supporting local neighborhood for $L_1$-medial skeleton construction.

In order to prevent the appearance of non-uniform distribution situations, such as point clusters, it is proposed to add a conditional regular term $R(X)$ to apply a repulsive force during the generation of local skeleton points, so as to avoid point offset due to iteration when the initial skeleton points extracted by the $L_1$-Median algorithm are already at the appropriate positions, and to ensure the uniform distribution of the initial skeleton points.

The classical weighted Principal Component Analysis (PCA) is used to detect the distribution of the point cloud near the individual tree branch skeleton structure. For a point $x_i$ in the point set $X$, the eigenvalues and eigenvectors of a $3 \times 3$ weighted covariance matrix are calculated as shown in Equation (3):

$$C_i = \sum_{i' \in I \setminus \{i\}} \theta(\parallel x_i - x_{i'} \parallel)(x_i - x_{i'})^T (x_i - x_{i'}). \tag{3}$$

To conditionally apply the repulsion force, we define our regularization function as in Equation (4) [27]:

$$R(X) = \sum_{i \in I} \gamma_i \sum_{i' \in I \setminus \{i\}} \frac{\theta(\parallel x_i - x_{i'} \parallel)}{\sigma_i \parallel x_i - x_{i'} \parallel} \,, \tag{4}$$

where $\gamma_i$ is the balancing constant of the optimal point set $X$, $\sigma_i$ is the directionality degree of $x_i$ within a local neighborhood, and its calculation formula is as in Equation (5):

$$\sigma_i = \sigma(x_i) = \frac{\breve{}_i^2}{\breve{}_i^0 + \breve{}_i^1 + \breve{}_i^2} \,, \tag{5}$$

where $\breve{}_i^0$, $\breve{}_i^1$, $\breve{}_i^2$ are the eigenvalues of point $x_i$, where $\breve{}_i^0 \leq \breve{}_i^1 \leq \breve{}_i^2$ forms an orthogonal system, which is the principal component of the point set. The closer $\sigma_i$ is to 1, the smaller

$\breve{}_i^1$ and $\breve{}_i^0$ are compared to $\breve{}_i^2$; and hence, the more points around $x_i$ are aligned along the direction of the tree skeleton.

After determining the regular term $R(X)$, this paper calculates the $L_1$-median point $x_i$, $\alpha_{ij} = \frac{\theta(\|x_i - q_j\|)}{\|x_i - q_j\|}$, $\beta_{ii'} = \frac{\theta(\|x_i - x_{i'}\|)}{\|x_i - x_{i'}\|^2}$. When the energy gradient value is 0, the fixed coefficient at each point should satisfy Equation (6) [27]:

$$\sum_{j \in J}(x_i - q_j)\alpha_{ij} - \gamma_i \sum_{i' \in I \setminus \{i\}} \frac{x_i - x_{i'}}{\sigma_i}\beta_{ii'} = 0. \tag{6}$$

At this point, the parameter $\mu$ is defined, and $\mu$ satisfies Equation (7) [27]:

$$\mu = \frac{\gamma_i \sum_{i' \in I \setminus \{i\}} \beta_{ii'}}{\sigma_i \sum_{j \in J} \alpha_{ij}}, \forall i \in I. \tag{7}$$

In order to avoid the $x_i$ coefficient matrix being singular, let $0 \leq \mu\sigma_i < 1/2$, and the $L_1$ median point $x_i$ is solved iteratively at the same time. Note, that the median point set in the current iterate $X^k = \{x_i^k\}$, $k = 0, 1, \cdots$, then the point set $\{x_i^{k+1}\}$ generated in the next iterate is as shown in Equation (8):

$$x_i^{k+1} = \frac{\sum_{j \in J} q_j \alpha_{ij}^k}{\sum_{j \in J} \alpha_{ij}^k} + \mu\sigma_i^k \frac{\sum_{i' \in I \setminus \{i\}} (x_i^k - x_{i'}^k)\beta_{ii'}^k}{\sum_{i' \in I \setminus \{i\}} \beta_{ii'}^k}, \tag{8}$$

where $\alpha_{ij}^k = \frac{\theta(\|x_i^k - q_j\|)}{\|x_i^k - q_j\|}$, $j \in J$; $\beta_{ii'}^k = \frac{\theta(\|x_i^k - x_{i'}^k\|)}{\|x_i^k - x_{i'}^k\|}$, $i' \in I \setminus \{i\}$; $\sigma_i^k = \sigma(x_i^k)$.

Since $\sigma_i^k \in (0, 1]$ can adaptively adjust the repulsive force based on the point dominant direction. In this paper, only the control parameter $0 \leq \mu < 1/2$, can control the penalty strength of the overall individual tree point cloud during the iterative shrinkage process.

This paper iteratively shrinks the set of sampling points according to the initial neighborhood value. The initial contraction radius is set according to the initial neighborhood radius, as shown in Equation (9) [22,27]:

$$h_0 = 2d_{bb}/\sqrt[3]{|J|}, \tag{9}$$

where $h_0$ is the initial neighborhood value, $d_{bb}$ is the diagonal length of the input $Q$'s bounding box and $|J|$ is the number of points in $Q$.

### 2.1.3. Searching Key Points in Skeletons

To identify branch points in the set of labeled sampling points, this paper calculates the directional metric $\sigma_i$ for all non-branch points and eliminates outlying points based on the $k$-nearest neighbor algorithm (the default $k$ value set is 6). Meanwhile, the corresponding threshold value is set for $\sigma_i$. Based on the experience of reference [22], it is set that when $\sigma_i > 0.9$, the point $x_i$ is identified as a candidate branch point, and it is determined that all points in the neighborhood of point $x_i$ have the same directional distribution at this time. Then the labeled branch points are further identified based on the candidate branch points. In this paper, the point corresponding to the maximum $\sigma$ value is labeled as seed point $x_0$, and all candidate branch points in its neighborhood are traversed from point $x_0$. Specifically, this paper calculates the distribution direction of point $x_i$ based on PCA and searches for candidate branch points in the vicinity of point $x_i$ along the direction. The search process for branch points is terminated when there are no points within the neighborhood that satisfy condition (10):

$$\cos\left(\angle\left(X_i\vec{X}_{i-1}, X_i\vec{X}_{i+1}\right)\right) \leq -0.9, (i = \cdots, -1, 0, 1, \cdots). \tag{10}$$

The new iteration process selects the largest $\sigma_i$ value from the remaining candidate branch points as the new seed point and iterates until all candidates have been processed.

In the process of searching for branch points, if a fixed value $h_0$ is used as the neighborhood radius for searching, there are often some branch points that are incorrectly marked as non-branch points, thus causing some regions of branch skeletons to be missing. Therefore, an adaptive neighborhood radius value $h$ is needed to better adapt to the variation of the different individual tree structures and avoid the phenomenon of over-shrinkage and under-shrinkage. Based on the assumption that the individual tree point cloud gradually becomes less dense along the skeleton structure from the root to the end of the branch, we gradually increase the size of the $h$ value during the shrinkage iterations, while eliminating the branch points that have been marked. At each iteration, the neighborhood value is to be increased by $\Delta h$, and the equations are as follows:

$$h_i = h_{i-1} + \Delta h, \tag{11}$$

$$\Delta h = h_0/4. \tag{12}$$

### 2.2. Local Feature Calculation of Point Cloud (Calculation of Dominant Direction and Local Point Cloud Weight Density)

Based on the extracted $L_1$-Median initial skeleton key points, the dominant direction and local point cloud weight density of skeleton key points in the individual tree point cloud were defined, and the adjacent correlation points of the missing area were guided to move along the dominant direction, and the individual tree incomplete point cloud was gradually repaired.

Specifically, based on the assumption that each branch extends in a unique direction, the extension direction of the branch to which each point belongs is defined as the dominant direction of the point, and the dominant direction of the initial skeleton point is used to represent the dominant direction of the points in the neighborhood of the skeleton point. Firstly, the $k$ nearest neighbors of the initial skeleton point $i$ are obtained based on the $k$-nearest neighbor algorithm [35], and the dominant direction of these $k$ nearest neighbors are defined to be the same as the dominant direction of the initial skeleton point $i$. The subnodes of the initial skeleton point can be divided into three cases: (1) contains only a single sub-node, i.e., this initial skeleton point is the internal point of the branch; (2) contains two or more sub-nodes, i.e., this initial skeleton point is the branch bifurcation point; (3) does not contain sub-nodes, i.e., it is a branch end skeleton point. When the initial skeleton point is located inside the branch, the dominant direction of the initial skeleton point $i$ is calculated schematically as shown in Figure 3a. Where the red nodes $i$, $j$, and $k$ represent the initial skeleton points, the yellow, blue, and green nodes represent the original point cloud (different colors represent the nearest neighbors of different skeleton points), and the green line represents the dominant direction of skeleton points $i$ and $j$. The dominant direction of the skeleton point is the direction that the initial skeleton point $i$ points to its unique child node $j$. This direction also represents the dominant direction of the $k$ points in the neighborhood of point $i$ (the points indicated in yellow in the figure). When the initial skeleton point is located at the branch bifurcation point, at this time, based on the single child node dominant direction calculation method, the dominant direction is calculated for each initial skeleton point of the branch route in turn, and the calculation schematic is shown in Figure 3b, where the red node is the initial skeleton point and the green line represents the dominant direction of the point. In the case of Figure 3b, this paper first calculates the dominant direction of all skeleton points on branching route a, and then calculates the dominant direction of all skeleton points on branching route b based on the bifurcated skeleton point $i$. When the initial skeleton point is located at the end of the branch, the traversal iteration ends at this point, and the dominant direction of the end skeleton point is set to be the same as the dominant direction of its parent node.

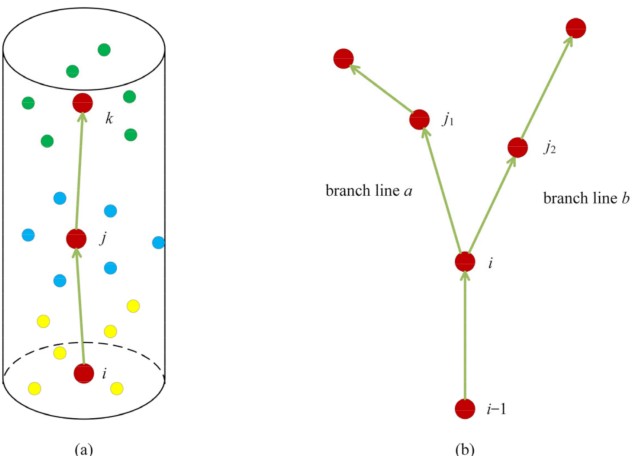

**Figure 3.** Calculation of the dominant direction of the initial skeleton point. (**a**) Single child node; (**b**) Multi child nodes.

After determining the dominant direction of the initial skeleton point, the point cloud is guided to move according to the dominant direction of the key point. Drawing on the idea of structure-aware global optimization [21], this paper analogizes point clouds as particles with electric power. Then these particles move freely according to the dominant direction, and the repair of the point cloud in the missing region is completed when the particles reach force equilibrium and stop moving. For the above purpose, it is assumed that each point in these discrete point clouds is a particle with the same kind of charge. They repel each other, so the particle at the end of the branch point cloud will be repelled by its forward particle and move. However, these particles cannot be moved arbitrarily, because the branches have directions, so it is necessary to obtain the direction of each branch, and then let these particles move along the direction of this branch extension. Next, it is necessary to define the force between the particles to control the range of motion of these particles, and $F_r$ is defined to express the repulsive force of each particle by the surrounding particles. The direction of this force follows the extension direction of the branch, which is the projection of the repulsive force of the surrounding particles on this particle in the extension direction of the branch, either in the same direction as the extension of the branch or in the opposite direction. However, if there is only a repulsive force, the particle at the end will always move in the direction of the branch extension and will not stop, so a binding force $F_s$ is needed to prevent the point from deviating significantly from the original position of the point. The direction of this force should be from the current position to the original position of the point. The farther the point deviates from its original position, the stronger the force should be, similar to a spring force. A particle is subjected to the joint action of these two forces, and when the particle finally stops moving, the two forces should be equal in magnitude and opposite in direction, so that the particle is in a state of force equilibrium. When all the particles stop moving, the missing part is repaired by these particles.

To quantify the force equilibrium state of the particles (point cloud), a constraint is imposed on the moving distance of the points, and the local point cloud weight density $d_j$ of the key points of the skeleton is defined, and the calculation formula is as in Equation (13):

$$d_j = 1/v_i, \tag{13}$$

where $v_i$ is the average distance of all points connected to the skeleton point $i$ of the key point.

The local point cloud weight density can ensure that the tree point cloud moves within a small range, following the vegetation growth law conditions. For example, in the area with a larger local point cloud weight density, it indicates that the point cloud is closer to its skeleton key point, the point cloud density is higher, and the distance to be moved

becomes larger accordingly. Accordingly, in the region where the local point cloud weights are less dense, the neighboring points are farther away from the skeleton key points, the point cloud is sparser, and the distance to be moved becomes smaller accordingly.

### 2.3. Iterative Repair Optimization

After obtaining the dominant direction and local weight density of the point, in order to further quantify the balancing condition, the idea of gravitational and repulsive force construction is borrowed from reference [21] to define the action force and contractive constraint of the method point in this paper, where the formula of the action force $F_r(i)$ is shown in Equation (14):

$$Fr(i) = \sum_{j \in \Omega_j} O_i^T \frac{d_i d_j}{\parallel P_i - P_j \parallel^2} (P_i - P_j) O_i, \tag{14}$$

where $d_i$ and $d_j$ are the local weight density of point $I$ and point $j$, $P_i$ and $P_j$ are, respectively, the space coordinates of point $i$ and point $j$ after iteration contraction, $O_i$ is the dominant direction of point $i$, and $T$ is the transpose operator.

The contractive constraint $F_s(i)$ can be calculated as Equation (15):

$$F_s(i) = \varphi(U_i - P_i), \tag{15}$$

where $U_i$ is the initial spatial position of point $i$, $\varphi = \frac{\sum j \in \Omega_j d_j d_i}{\frac{1}{m} \sum_1^m v_i} \log_2(c_i + 1)_i$, which is an adjustment factor to prevent the point cloud at the end of the branch from moving widely due to low local weight density, so the $\varphi$ value will be larger at the missing location and smaller at the end of the branch. $m$ is the number of all $L_1$-Median skeleton key points, and $c_i$ is the sum of Euclidean distances of all child nodes of point $i$ in the $L_1$-Meidan skeleton structure.

Through the above optimization process, the incomplete individual tree point cloud is iteratively repaired and optimized. When the point force and contraction constraints reach balance, the iterative process terminates and the incomplete point cloud is repaired.

### 2.4. Repair Effect Evaluation

To further evaluate the restoration effect of tree point clouds, this paper quantitatively evaluates the point cloud repair optimization results with the help of the final modeling effect based on the quantitative analysis model in the literature [36]. Specifically, an individual tree with good point cloud integrity was selected as the study object, and a 3D tree model was constructed based on this point cloud data, which was recorded as the reference individual tree model. Then, some branch point clouds were manually removed, and the point cloud data of the individual tree after each iteration was output through the above iterative restoration process. To quantify the repair effect of each iteration, the individual tree point cloud generated after each iteration was constructed as a 3D individual tree model and recorded as a validation individual tree model. For the constructed 3D model, the TLS point cloud generation process was simulated: specifically, the laser beam of the 3D laser scan was simulated using PBRT (Physically-Based Ray Tracing) software [37], and the field of view was set to the same value as that of the scanner to obtain the point cloud, which was set to $-60°-90°$. The reconstructed individual tree model was placed at a distance so that the simulated beam could scan the entire tree. Three laser scans were simulated at azimuths of $0°$, $120°$, and $240°$ around the individual tree, and the distance between the camera and the tree model was kept constant. Each simulated scan produces a raster image. For each pixel of the raster image, the simulated scan emits a laser beam to ensure the relative integrity of the simulated point cloud generation, and finally, the simulated point cloud is aligned to generate a 3D point cloud of an individual tree. For the simulated generated individual tree point cloud, this paper continues to compare the differences between the point clouds in the two forms to verify the differences between the

reference individual tree model and the validated individual tree model. In other words, the simulated individual tree point cloud is rasterized and a 3D raster (voxel) of size 0.2 m is created in the point cloud space enclosing the box, and then the simulated individual tree point cloud is put into the voxel and the number of points in each voxel is recorded. Finally, the number of simulated transformed points in the voxels with points is counted to obtain the difference between the individual tree model to be validated and the reference individual tree model. To facilitate comparison, the differences between the number of points are further normalized and the mean and standard deviation of these differences are counted.

### 3. Results and Discussion

#### 3.1. Experimental Data

In order to verify the reliability and effectiveness of the point cloud restoration algorithm proposed in this paper, Leica C10 and Faro X330 3D laser scanners were used to obtain part of individual tree point cloud data for experimental analysis. In order to reduce the influence of leaves on the acquisition of branch point clouds, all individual tree point clouds in the experiment were collected in winter or early spring, and the trees contained only a few leaves. In order to test the above algorithm, the individual tree data with obvious branch point cloud missing were selected for experimental analysis. Table 1 lists the relevant information of three individual trees used in the point cloud repair optimization experiment, including tree species, scanner adopted, scanning distance and the number of points of an individual tree point cloud.

**Table 1.** The information of experiment individual tree.

| Tree Species | *Ginkgo biloba* | *Platanus acerifolia* | *Cerasus serrulata* | *Robinia pseudoacacia* |
|---|---|---|---|---|
| Scanner | Faro X330 | Leica C10 | Leica C10 | Faro X330 |
| Scanning distance/m | 8 | 22 | 16 | 5 |
| Scanning mode | Single station | Single station | Single station | Single station |
| Number of points | 265,428 | 24,326 | 30,804 | 228,398 |

#### 3.2. Individual Tree Initial Skeleton Extraction

In order to explore the appropriate number of sampling points, this paper conducts experiments and analyses on a different number of sampling points *N*. Figure 4 shows the effect of extracted $L_1$-Median initial skeleton points of *Cerasus serrulata* tree with a different number of sampling points. Among them, Figure 4a shows the original individual tree point cloud of the obtained *Cerasus serrulata* tree. Figure 4b–f shows the initial skeleton of *Cerasus serrulata* tree extracted at 5%, 10%, 15%, 20% and 25% of the total number of sampling points, respectively. Moreover, it can be seen that the skeleton structure of the extracted generated *Cerasus serrulata* tree is gradually improved with the increase of sampling proportion, while the skeleton structure changes less when the proportion of sampling points exceeds 20%, and at this time, with the increase of the number of sampling points, the time cost of extracting the initial skeleton points increased significantly. Therefore, the sampling points with 20% of the number of individual tree point clouds were finally selected and marked as non-branching points to participate in the subsequent iterative shrinkage calculation.

To determine the appropriate *μ* value, the $L_1$-Median initial skeleton point extraction with different *μ* values is tested in this paper. Figure 5 shows the effect of $L_1$-Median initial skeleton points extracted under different values of penalty strength *μ*. When the value is less than 0.3, more twig skeleton points are not effectively extracted, and the extracted initial skeleton points can only abstract the trunk and branches with a thicker radius, which will obviously affect the subsequent local point cloud feature calculation and thin branch incomplete point cloud repair. When the value of *μ* is greater than 0.35, the difference between the initial skeleton points of individual tree extracted based on $L_1$-Median extraction and those extracted with *μ* = 0.35 is small. However, with the same

settings of other parameters, the number of iterations and time complexity of the algorithm increases significantly as the value of $\mu$ increases. For example, the time spent to extract the skeleton points under $\mu = 0.4$ is nearly twice the time spent under $\mu = 0.35$, and the number of iterations is significantly higher: under $\mu = 0.35$, the iteration shrinks 87 times, while under $\mu = 0.4$, the iteration shrinks 174 times. Moreover, as can be seen from Figure 5, the initial skeleton point locations are similar for the two threshold settings. Therefore, $\mu = 0.35$ was finally selected as the default threshold for the $L_1$-Median algorithm with conditional regularization term.

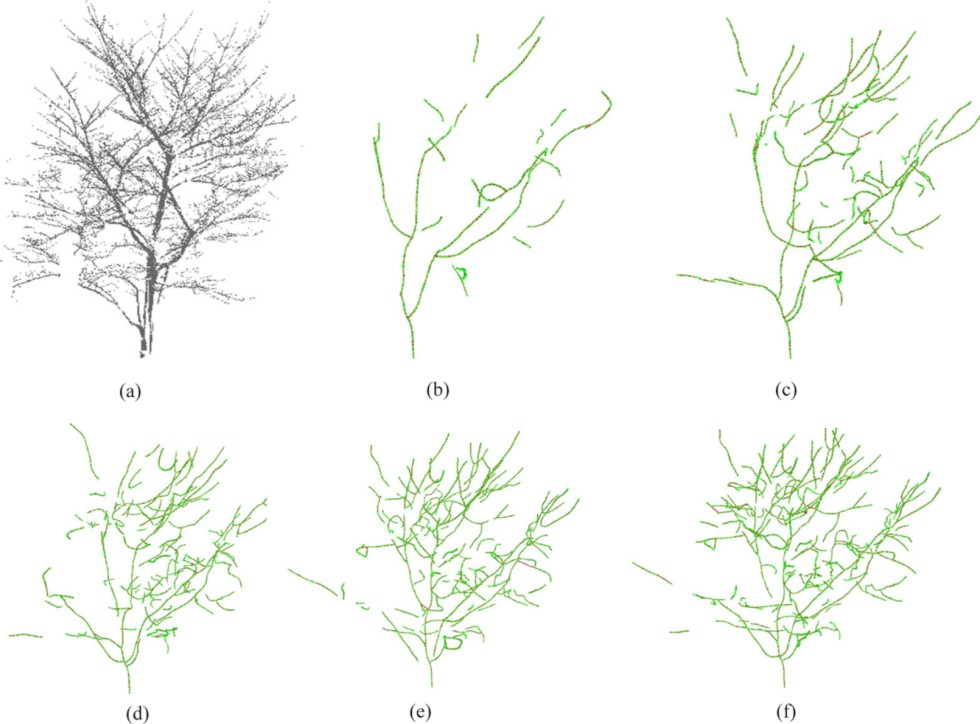

**Figure 4.** $L_1$-Median initial skeleton points extracted at different sampling points. (**a**) Initial point clouds; (**b**) *N*–5%; (**c**) *N*–10%; (**d**) *N*–15%; (**e**) *N*–20%; (**f**) *N*–25%.

Three typical individual trees with different geometric and topological features were selected: *Ginkgo biloba*, *Platanus acerifolia* and *Cerasus serrulata*. Their initial skeletons were extracted using the method described in Section 2.1, and the initial parameter settings of skeleton point extraction were shown in Table 2. Figure 6 shows the original point clouds of *Ginkgo biloba*, *Platanus acerifolia* and *Cerasus serrulata*, respectively. Part (b) is the extracted initial skeleton point and skeleton structure, where the red points are the initial skeleton points and the green lines are the skeleton lines. In addition, the part in the red frame in the figure is the missing part of the point cloud caused by external factors.

**Table 2.** The initial parameter setting of three individual tree initial skeleton point extraction.

| Parameter<br>Tree Species | Sampling Points | - | $h_0$ | $\Delta h$ | $K$ |
|---|---|---|---|---|---|
| Initial value | 1000 | 0.35 | $2d_{bb} / \sqrt[3]{\lvert J \rvert}$ | $h_0/4$ | 6 |
| *Ginkgo biloba* | 20,000 | 0.35 | 0.1707 | 0.043 | 8 |
| *Platanus acerifolia* | 5000 | 0.35 | 0.8992 | 0.2248 | 6 |
| *Cerasus serrulata* | 6000 | 0.35 | 0.3996 | 0.0999 | 6 |

Note: $\mu$ is the parameter defined in Equation (6), $h_0$ is the initial neighborhood value, $d_{bb}$ is the diagonal length of the input $Q$'s bounding box and $\lvert J \rvert$ is the number of points in $Q$, $Q$ is the original input individual tree point set. $\Delta h$ is the decreasing value of $h$ in each iteration. $K$ is the number of points selected by *k*-nearest neighbor algorithm.

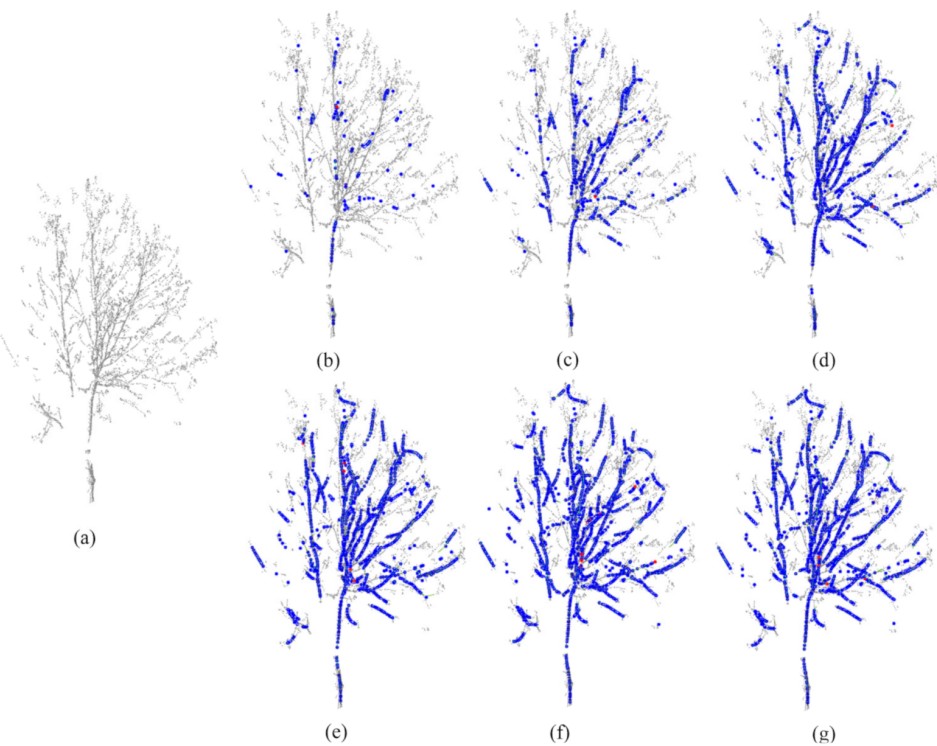

**Figure 5.** The initial skeleton points extracted under different penalty strength $\mu$ values, the blue points are the initial skeleton points, and the gray points are the original point cloud. (**a**) Initial point clouds; (**b**) $\mu = 0$; (**c**) $\mu = 0.1$; (**d**) $\mu = 0.2$; (**e**) $\mu = 0.3$; (**f**) $\mu = 0.35$; (**g**) $\mu = 0.4$.

In order to facilitate the selection and calculation of branch points, the sampling points in the test were rounded according to the calculation method of sampling points in this section.

The experimental results show that the $L_1$-Median initial skeleton point extraction algorithm has the ability to deal with trees with different geometric structures. Figure 6a,c shows that this algorithm has good applicability to tree structures of different complexity. Figure 6b shows a *Platanus acerifolia* with the missing trunk point cloud, sparse canopy branch point cloud and a little noise. Although the $L_1$-Median algorithm can obtain the approximate skeleton point position, the incomplete point cloud also greatly affects the topological connection between skeleton points. As shown in the red frame, the skeleton point connection appears with different degrees of fracture (Figure 6a right part) and wrong connection (Figure 6b, right part) at the point cloud loss. Compared with the traditional skeleton point extraction algorithm, the $L_1$-Median algorithm has good robustness to input point cloud quality and good applicability to an individual tree with different geometric structures. Compared with the graph theoretic methods, such as Minimum Spanning Tree (MST), which extract skeleton points mostly along the individual tree surface, the $L_1$-Median algorithm can directly obtain points that approximate the center of the local individual tree point cloud. Although the skeleton point extraction process loses some time efficiency, it ensures that the initial skeleton points are distributed roughly along the extension direction of the central axis of the tree branches. It is beneficial to the local feature calculation of point clouds and the research of incomplete point cloud repair in this chapter.

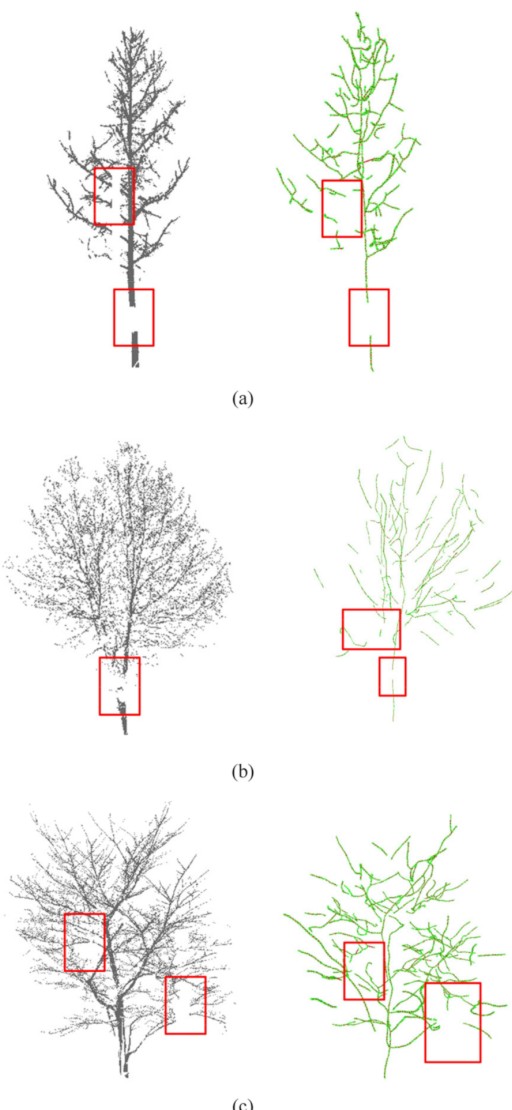

**Figure 6.** The initial skeleton points extracted by L$_1$-Median algorithm. (**a**) *Ginkgo biloba*; (**b**) *Platanus acerifolia*; (**c**) *Cerasus serrulate*. The left column is the original point cloud, the right column is the initial skeleton structure. The red frame is the incomplete point cloud caused by the blocking of street lights and vehicles, the red dots are the skeleton points, and the green lines are the skeleton lines.

In order to verify that the L$_1$-Median algorithm can deal with the arbitrary change of point cloud density, the point cloud of the *Cerasus serrulata* tree with missing data was randomly thinned and the initial skeleton points were extracted by the thinned points. Figure 7 shows the extraction results of skeleton points under the conditions of 100%, 70%, 50% and 30% dilution of the original point cloud. As can be seen from Figure 7, with the decrease of point cloud density, the backbone skeleton structure is still well maintained. Because the main skeleton structure of the *Cerasus serrulata* tree is relatively similar. After thinning treatment, the removal of a large number of twig point clouds affects the connection performance between skeleton points (red frame content), resulting in a change of topological connection relationship in some regions, but the overall structure is similar. The results show that the L$_1$-Median initial skeleton point extraction algorithm is robust to point density changes.

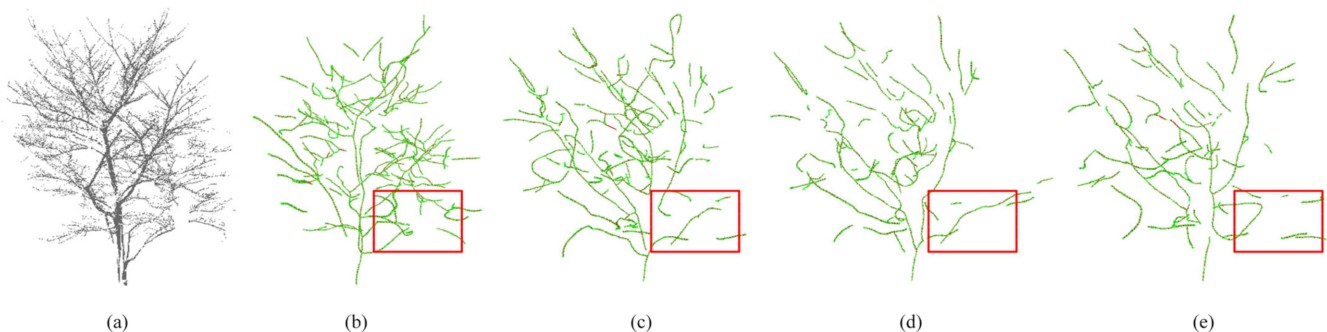

**Figure 7.** The reconstructed tree models using the different density of point clouds. (**a**) Initial point clouds; (**b**) 100%; (**c**) 70%; (**d**) 50%; (**e**) 30%.

### *3.3. Individual Tree Point Cloud Repair*
#### 3.3.1. Qualitative Evaluation

The point cloud data of *Ginkgo biloba*, *Platanus acerifolia* and *Cerasus serrulata* trees with large data missing in the previous section are adopted, and the point cloud spatial location optimization and iterative repair algorithm are used to repair and optimize the incomplete point cloud. Figure 8 shows the point cloud restoration results of *Ginkgo biloba*, Cerasus serrulate and *Platanus acerifolia*. Among the three individual trees, the ginkgo tree point cloud density was large, but due to the shielding of street lights and billboards, part of the branch point cloud was missing and showed regular strips. The point cloud and skeleton topology of the missing parts could be well restored based on the point cloud repair optimization algorithm. The point cloud skeleton of *Cerasus serrulata* trees was relatively clear, but the point cloud density was low. Optimization and iterative restoration could recover some of the branch point clouds at the fracture, but the effect of point cloud restoration was not obvious due to the sparse point cloud of canopy twigs, and the skeleton lines of some twigs could not be accurately extracted. The quality of the initial point cloud of *Platanus acerifolia* was poor, the position of the initial skeleton points obtained deviated, and the skeleton structure was poor. After optimization and iteration repair, the overall skeleton structure was well maintained, and some incomplete point clouds were restored to a certain extent, but the deviation of the initial skeleton points also caused the calculation error of the dominant direction of the iteration. Therefore, part of the point cloud moved in the wrong direction, resulting in great differences between the skeleton structures. The experimental results shown in Figure 8 show that the iterative point cloud repair algorithm proposed in this paper can effectively deal with a small number of incomplete point clouds and recover the point cloud data of some missing branches. For trees with a clear skeleton structure, the location of adjacent point clouds can be adjusted and iteratively optimized to make the optimized tree point cloud density more uniform than before optimization, and the method has good robustness.

In order to further verify the robustness of the iterative repair algorithm for the incomplete point clouds, a *Robinia pseudoacacia* tree with good point cloud integrity was selected to manually delete part of the branch point cloud of the individual tree, and then compare the individual tree skeleton before and after the deletion of the point cloud. In this section, the 4.4 m–4.7 m area of *Robinia pseudoacacia* canopy was selected for manual point cloud deletion, which has more branches and more complex geometry inside the area. After deleting the point cloud, a 0.3 m-wide strip point cloud was missing from the original *Robinia pseudoacacia* individual tree. Figures 9a and 9d, respectively, represent the original tree point cloud and the tree point cloud after removing some branch points, and Figure 9b,c and Figure 9e,f, respectively, represent the extracted skeleton structure based on the original tree point cloud and the tree point cloud after deleting points. From the content of the red frame in Figure 9f, it can be seen that after deleting the points, the branch skeleton topology shows obvious connection errors due to the incomplete point

cloud. Figure 9g is the optimized tree point cloud, and Figure 9h,i is the extracted skeleton structure, comparing Figure 9a,c with Figure 9g,i shows that after the point cloud iterative repair optimization algorithm, most of the point clouds of branches are effectively repaired. The skeleton structure of the individual tree main branch obtained before deletion and after repair and optimization is similar, while some twigs are deleted together with the over-sparse point cloud in the process of deletion so that the subsequent optimization process is not successfully repaired. The similar skeleton before and after point deletion shows that the iterative repair optimization algorithm in this paper can effectively deal with the incomplete point clouds and recover the point cloud data at the missing branches. Moreover, at the same time, the optimized point cloud is used to construct the tree skeleton to approximate the skeleton structure under the complete point cloud. The repaired and optimized individual tree point cloud can effectively abstract the real skeleton structure and realize the high-precision reconstruction of tree mode.

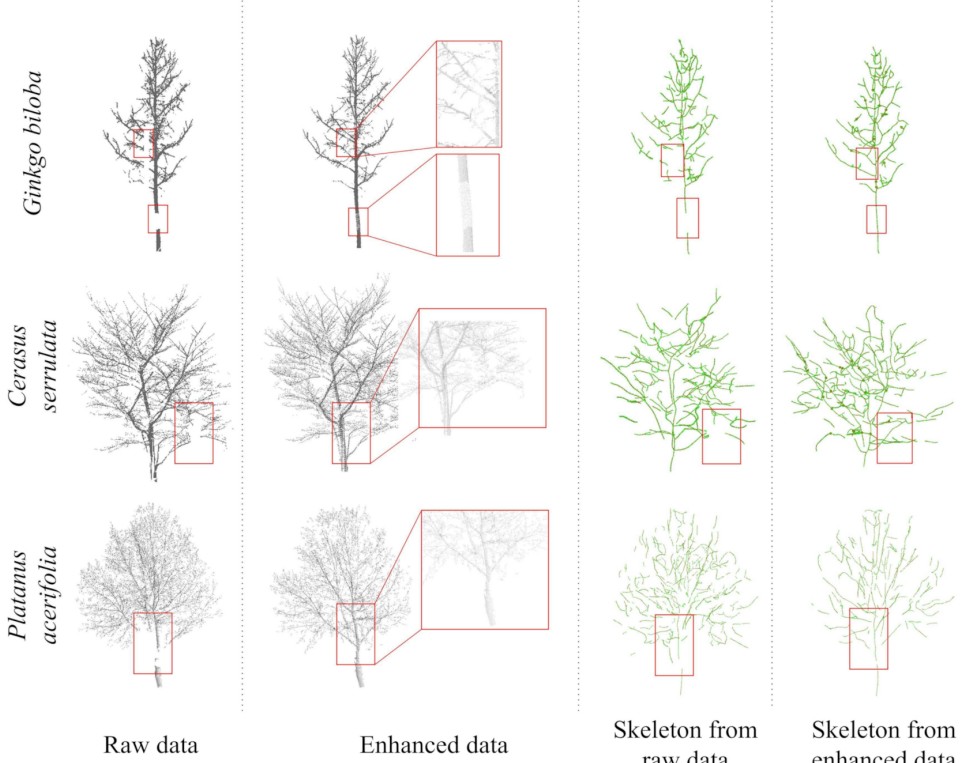

**Figure 8.** Schematic diagram of point cloud restoration of *Ginkgo biloba*, *Cerasus serrulata* and *Platanus acerifolia*.

### 3.3.2. Quantitative Evaluation

To further evaluate the recovery effect of tree point clouds, this paper constructs a reference individual tree model based on *Robinia pseudoacacia* tree point clouds, and a validated individual tree model based on the point clouds generated by each iteration of optimization after manually deleting some branch point clouds. Figure 10 describes the differences between the individual tree model to be verified constructed by the optimization point cloud of the first, second and last iterations and the reference individual tree model. The mean and standard deviation of the differences between individual tree model to be validated and reference model are shown in Table 3.

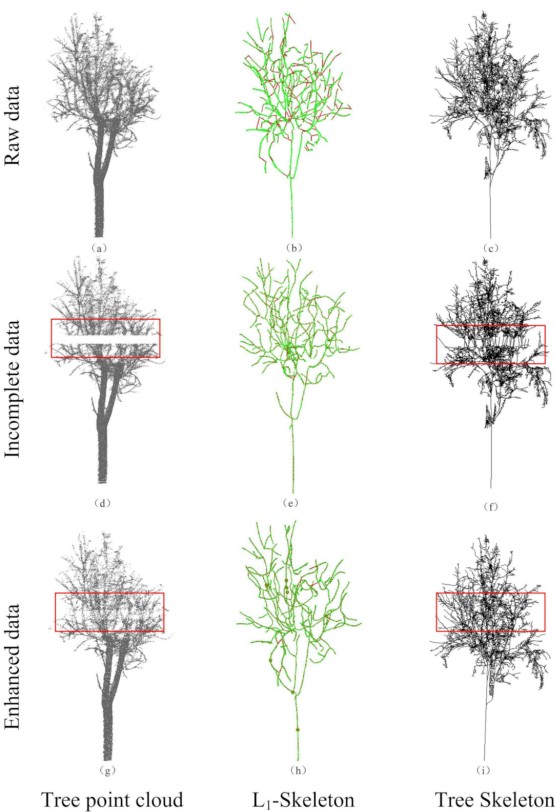

**Figure 9.** Comparison of *Robinia pseudoacacia* tree point clouds before and after deleting points. (**a**) is the original point cloud, (**b**,**c**) is the extracted skeleton structure; (**d**) describes the tree point cloud after deleting some points, (**e**,**f**) is the extracted skeleton structure; (**g**) represents the tree point cloud after iterative repair, (**h**,**i**) is the extracted skeleton structure.

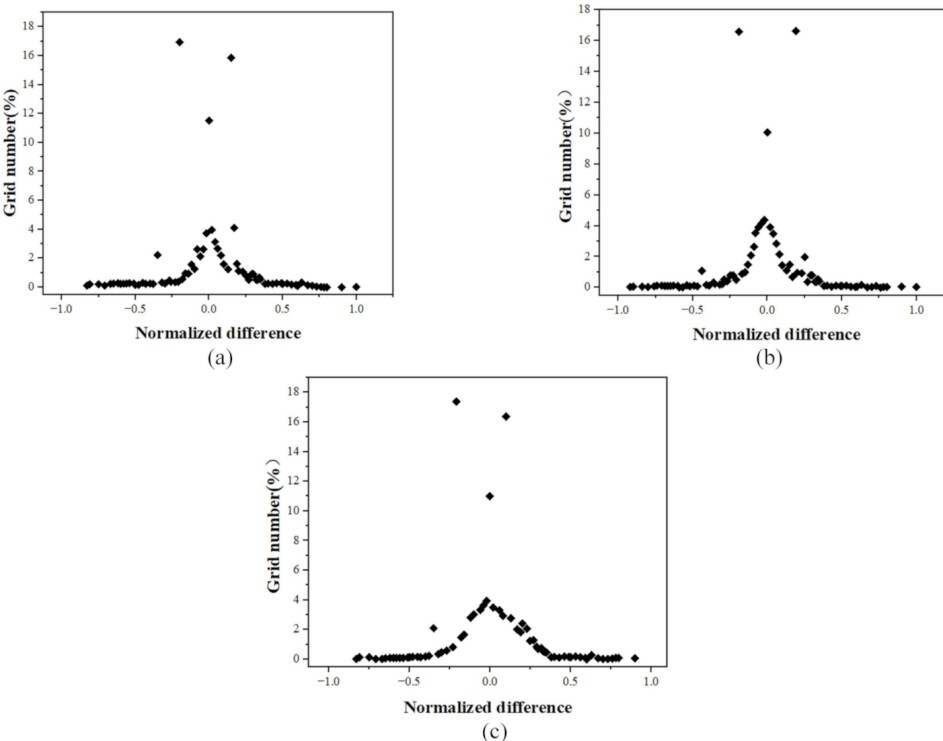

**Figure 10.** Point cloud distribution difference between the reconstructed model in each iteration and the reference individual tree model. (**a**) First iteration; (**b**) Second iteration; (**c**) Last iteration.

**Table 3.** Differences between individual tree models to be validated and reference models.

|  | First Iteration | Second Iteration | Last Iteration |
|---|---|---|---|
| Mean (m) | −0.0067 | −0.0054 | −0.0036 |
| Standard Deviation (m) | ±0.189 | ±0.176 | ±0.167 |

In order to verify the skeleton reconstruction accuracy after branch repair in the incomplete point cloud area, this paper used the repair-optimized individual tree point cloud, reconstructed its skeleton structure and calculated the radius of the skeleton point to represent the radius of the branch at that skeleton point. The results are shown in Table 4. It can be seen from the results that for the four restored individual trees, the branches with a clear structure and dense point clouds were basically restored effectively, while the incomplete point clouds were poorly restored in the branches with sparse point clouds and small radii. In addition, due to the high position of some tree branches, it is difficult to accurately measure their branch radius. In the experiment, only the branches near the main trunk with missing points were used as the experimental objects, and their radii were determined by measuring the branch circumferences several times. From the radius comparison results, it can be seen that the radius of the repaired branch is smaller than that of the original branch. This is because the neighboring points in the missing area move towards the missing area during the point cloud restoration process, which leads to the gentle change of point cloud density in the local area of the branch. Moreover, the density of the restored point cloud is lower than that of the scanned point cloud, which eventually causes the radius of the restored branch to be smaller than that of the real tree.

**Table 4.** Branch accuracy after repair and optimization.

| Tree Species | Number of Branches | | Branch Radius/cm | |
|---|---|---|---|---|
|  | Missing Branches | Repaired Branches | True Branch | Repaired Branch |
| *Cerasus serrulata* | 46 | 27 | 2.2 | 1.9 |
| *Platanus acerifolia* | 19 | 15 | 9.4 | 8.8 |
| *Robinia pseudoacacia* | 33 | 29 | 5.3 | 4.6 |

### 3.3.3. Comparison with Other Methods

To further illustrate the effectiveness of the algorithm in this paper, the *Ginkgo biloba* point cloud data used in reference [21] were used to compare the effect of the point clouds restored by the algorithm in this paper, the structure-aware global optimization algorithm [21] and the joint modeling algorithm [38]. Considering that it is difficult to analyze the advantages and disadvantages of the three methods directly based on the restored tree point clouds from a subjective vision and that the structure-aware global optimization algorithm and the joint modeling algorithm do not show intermediate skeleton point outputs. Therefore, the point cloud iteratively restored skeleton points were initially topologically connected and the initial skeleton structure of the tree was constructed to facilitate the comparison of the three methods. Figure 11 shows the skeleton point extraction results for the *Ginkgo biloba* point cloud with a part missing under the three methods. The content of the red frame in Figure 11 shows the enlarged geometric details of the branch skeleton, from which the differences in details of the three methods in reconstructing the skeleton based on the restored point cloud can be seen. Livny's joint modeling algorithm reconstructs the branch point cloud directly based on the point cloud of the overlapping part of the tree, but due to the lack of fine segmentation, the branch point clouds of other individual trees also participate in the skeleton construction, so there are some extra branch skeletons that do not belong to the tree. Wang's structure-aware global optimization algorithm can recover the branch point clouds with large missing data, and the reconstruction process retains more thin branch skeletons and global features, but a little skeleton extension and topological connections have errors. The algorithm in this

paper can recover the incomplete point clouds of individual trees with clear branch features, but some twig point clouds are not recovered and optimized.

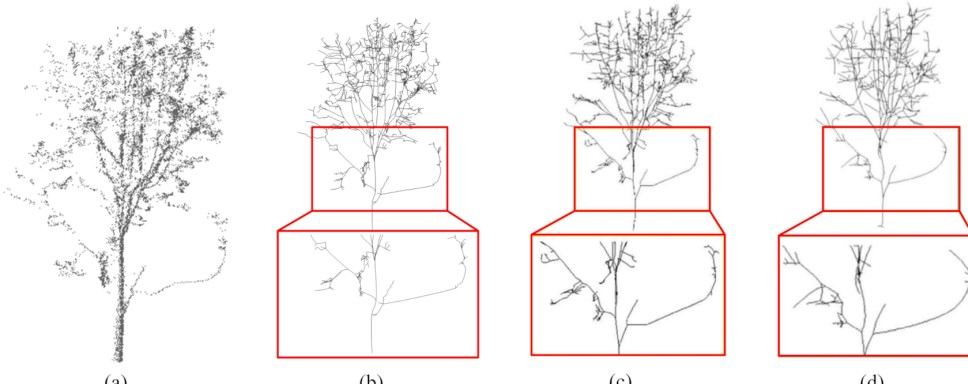

|     |     |     |     |
| --- | --- | --- | --- |
| (a) | (b) | (c) | (d) |

**Figure 11.** Comparison of *Ginkgo biloba* tree point cloud restoration and skeleton extraction results. (**a**) Tree point clouds; (**b**) Our method; (**c**) The method of Livny; (**d**) The method of Wang.

In terms of iterative repair, this paper focuses on extracting the balance method of the point force and contraction constraints, so as to repair incomplete point cloud. The algorithm is able to recover part of the incomplete point cloud of branches well for trees with clear skeleton structures. The tree skeleton constructed by using the repair-optimized individual tree point cloud is consistent with the real skeleton structure and can achieve high precision modeling of the tree model, but it is slightly weaker than the other two algorithms in the optimization of canopy twig detail repair.

## 4. Conclusions

To address the problem of missing partial point clouds in the field of three-dimensional point clouds of individual trees, this paper proposes an iterative restoration optimization algorithm based on local features of point clouds. The research focuses on the extraction of key points of individual tree skeletons from point clouds, the calculation of local features, and iterative restoration optimization. For individual trees with clear skeleton structures and simple branch geometry, the algorithm is able to extract more accurate skeleton key point locations and the restoration optimization effect is similar to that expected. However, for individual trees with complex crown branch structures, the point cloud quality at the fine branch level is poor and the restoration effect is lacking. The next step of the study is to consider coupling multi-source data with a priori knowledge of vegetation growth rule constraints to explore restoration methods for fine branches and branch ends.

**Author Contributions:** Conceptualization, Y.S. and D.C.; methodology, W.C. and D.C.; data curation, W.C. and J.W.; writing—original draft preparation, J.W. and W.C.; writing—review and editing, Y.S. and J.W. All authors have read and agreed to the published version of the manuscript.

**Funding:** This work was supported in part by the National Natural Science Foundation of China under Grant 41971415, and in part by the Natural Science Foundation of Jiangsu Province under Grant BK20201387. This work was performed while the co-author Dr. Dong Chen acted as an awardee of the 2021 Qing-lan Project, sponsored by Jiangsu Province, China.

**Institutional Review Board Statement:** Not applicable.

**Informed Consent Statement:** Not applicable.

**Data Availability Statement:** The data presented in this study are available on request from the corresponding author.

**Conflicts of Interest:** The authors declare no conflict of interest.

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
