# Peer review of "Restoration of Individual Tree Missing Point Cloud Based on Local Features of Point Cloud"

_remotesensing, doi:10.3390/rs14061346_

Round 1

Reviewer 1 Report

Some data missing parts often exist in the field-scan tree point clouds. The authors' work is novel and applicable for forest inventory and research on canopy phenotypic characteristics. They detailed introduced their methods. However, evaluating point cloud restoration accuracy should be added. Although it is difficult to evaluate the accuracy of skeleton extraction and data restoration for measured point clouds, it is feasible to quantitatively evaluate the processing accuracy of some simulated tree point clouds. I think the major issue is that many sentences in the current edition are redundant and lengthy, which decreases the manuscript's readability. And some logical confusion exists in the methods and results. I have pointed out some issues. I hope the authors carefully revise their manuscript and submit the post-review edition for rechecking before publication.

  1. In the introduction section, some sentences need to be rewritten to make them clearer.

“Three-dimensional information of vegetation such as trees is an important part of digital forestry process” (Line 23). “The absence of the regional point cloud usually adversely affects the extraction of vegetation parameters and the abstract expression of branches.” (Line 59).

  1. Most TLS datasets were not used in large operation scenes but at a plot level. (Line 30)
  2. “Leaf area index” is redundantly used on lines 36 and 38.
  3. Lines 71-97 The case shown in this part is too detailed to be described in the introduction.
  4. Lines 108-144 Authors are familiar with relevant point cloud processing algorithms. I suggest you summarize the related algorithms with the same characteristics together, not list them tip by tip. We need to summarize existing problems of existing algorithm and highlight the necessity of your proposed method.
  5. Lines 152-159 Some details about your algorithm need to be moved to section 2.
  6. Review about existing skeleton extraction methods are not appropriate show in Section 2. In my opinion, you can add the Background section to introduce them in Section 2.
  7. I think some discussion about the L1-Median algorithm (Lines 204-213) should not be described in Section 2. Please move them to the Discussion section.
  8. Lines 253-267 This part and Figure 3 show the penalty strength determination results. Moving it to the Results section (Section 3) would be better.
  9. Lines 271-294 Sample points selection should be before running the L1-Median algorithm. It would be better to change the methods introduction order in Section 2.
  10. Line 359 “particles with electric power”? I did not understand.
  11. In Eq.10, di and dj mean?
  12. Line 384 I did not understand this equation.
  13. Lines 401-408 The complexity of tree structure shown in point clouds cannot be set as an objective standard to distinguish the types of the point clouds, as it is affected by the scanning mode.
  14. In Table 2, some abbreviations need to be explained, although they have been described in Section 2. I'm not sure if it's necessary to use four decimal places. Moreover, the name of tree species in the parameter column is not suitable.
  15. Line 426 the yellow lines are the skeleton lines?I suggest combining Figures 6 to 8 into one image.
  16. Lines 463-468, 501-513 The description of some figures need not be too detailed.
  17. Line 495 A Robinia pseudoacacia tree? I did not find it in Section 3.1.
  18. In Fig 10, I found some initial skeletons were inaccurate. The extraction quality of skeletons is essential for the point cloud restoration.
  19. Lines 536-553 I think this part should move to section 2 to describe the accuracy evaluation methods of point cloud restoration.
  20. Line 542 What is the radiative transfer model? This paragraph is too long. You need to adjust some sentences to show your results more concisely.
  21. Line 546 “The mean difference of the individual tree models to be verified in the three iterations is less than 0.” This sentence is not clear.
  22. In Table 3, what are the units of these numbers?
  23. It would be better to put the first paragraph of Section 4 to Section 3, and rename Section 3 as “Results and Discussion”.

Reviewer 2 Report

Title: Please remove the term “Research on”

Authors: the surname start with capital letters such as “Cao”, please adjust them.

Abstract

The abbreviation LiDAR (Light Detection and Ranging) must be LiDAR (Light Detection And Ranging).

The abstract must consist of four parts: the goal of the suggested approach, method summarising, result and accuracy. Please rewrite the abstract and focus on these main elements (the abstract does not explain the paper motivation).

 Keywords:

 Keywords start with capital letters, LiDAR is a good keyword

 Introduction

It is better to use “Terrestrial Laser Scanning” (TLS) instead of “ground LiDAR”.

Line 102: Please replace the word “scholars” with “authors”. Please check all text, don’t use the words “researchers” “scientists” and “scholars”.

Line 104: Please cite directly references for each category.

Line 126: the expression “repaired the defects” is not suitable for academic papers. You can say “add improvements”.

Don’t use the word “defect”, you can use “disadvantage”.

The novelty of the paper has to be underlined.

The introduction section is merged with the related work section which is why it becomes so long and boring. Please separate it into two sections.

From the reference list, the newest reference was published in 2018 (only one paper in 2018) (for four years). During the last four years, a lot of contributions were published in the field of LiDAR data processing in green areas. The authors must update the reference list which can help them to improve the suggested approach regarding the modern published papers in similar research areas. Conclusion: the paper is outdating.

In the paper title, it is said: “Local Features of Point Cloud”. This expression is widely used during the last four years in the domain of LiDAR data processing among both machine learning and rule-based approaches. In the related work section, the authors have to add a subsection that summarised the widely used cloud features and a comparison between the machine learning and rule-based regarding the employed feature viewpoint. Moreover, the authors must clarify the local feature calculation approaches.

Some useful papers can help the authors to start updating their work:

Camuffo, E.; Mari, D.; Milani, S. Recent Advancements in Learning Algorithms for Point

Clouds: An Updated Overview. Sensors 2022, 22, 1357. https://doi.org/10.3390/s22041357.

Dey, E.K., Tarsha Kurdi, F, Awrangjeb, M., Stantic, B., 2021. Effective selection of variable point neighbourhood for feature point extraction from aerial building point cloud data. Remote Sens. 2021, 13,1520. https://doi.org/10.3390/rs13081520.

Tarsha Kurdi, F., Gharineiat, Z., Campbell, G., Awrangjeb, M., Dey, E.K., 2022. Automatic filtering of LiDAR building point cloud in case of trees associated to building roof, Remote Sens. 2022, 14, 430, https://doi.org/10.3390/rs14020430.

Chen, J.; Chen, Y.; Liu, Z. Classification of Typical Tree Species in Laser Point Cloud Based on Deep Learning. Remote Sens. 2021, 13, 4750. https://doi.org/10.3390/rs13234750.

Tarsha Kurdi, F., Gharineiat, Z., Campbell, G., Dey, E.K., Awrangjeb, M., 2021. Full series algorithm of automatic building extraction and modelling from LiDAR data. 2021 Digital Image Computing: Techniques and Applications (DICTA), pp. 1-8, doi:  10.1109/DICTA52665.2021.9647313, Gold Coast, Australia.

  1. Luo et al., "Detection of individual trees in UAV LiDAR point clouds using a deep learning framework based on multi-channel representation,2022, " in IEEE Transactions on Geoscience and Remote Sensing, doi: 10.1109/TGRS.2021.3130725.

Tree point cloud restoration

Figure 2 illustrates the workflow of the suggested approach. It consists of 6 consecutive steps; each step merits an independent subsection to explain it clearly.

The abbreviation “minimum spanning tree (MST)” must be written as “Minimum Spanning Tree (MST)” please check all abbreviations.

In the introduction (related work Line126 and 127) you said: “Mei et al. [17] repaired the defects based on the features of local point cloud coverage and point cloud density, but this method also had the defect of high time cost.” In line 238 you said “local neighborhood, and its calculation formula is as ….”. I see that there is a contradiction. You criticise the employment of local feature employment because that make high time cost, but you use it among the suggested approach to calculate the eigenvalues.

How do you detect the neighbouring points for calculating the eigenvalues?

What are the advantages of the suggested approach in regard to the literature?

Quantitative evaluation

How do you obtain the reference model (Line 540)? Is it really enough accurate to consider it as a reference?

Discussion and conclusion

Line 629: don’t use “etc”. Please check all the text.

Line 631: you said, “this algorithm increases the time complexity, but it can obtain more accurate individual tree internal points rather than surface skeleton points.” You criticise some methods (in the introduction section) because of the time cost, but you don’t solve this problem.

Round 2

Reviewer 1 Report

For sentences modified in the Introduction section, please pay attention to all conjunctions(such as Lines 121-123, and/or). Try to reduce the number of verbs in a sentence to avoid ambiguity (such as Lines 315-319).

Line 333 What are horizontal data sets?

Line 335 the more representative ones?

Please rename Section 2.1.2 as “Point cloud skeleton extraction”

Lines 401-403 Please rewrite this sentence.

Lines 403,406 What is J and I?

Please renamed Section 2.1.3 as “Searching key points in skeletons”.

Line 784 this paper->we

Line 785 iteration neighborhood value? There are two equations in Eq(11).

Lines 842-844 Please rewrite this sentence. Not clear.

Line 919 Add the reference of PBRT software.

Line 921 delete “in this paper”

All Latin names should be italicized.

Line 1018 What is Q?

Line 1086 shortest path?

Lines 1114-1117 Please delete this sentence.

Line 1156 Please check the description of Figure 8. You can mark the title of insets in this figure (Figure 9 show the same issue).

Lines 1366-1369 Please rewrite this sentence.

Line 1382 under->from

Reviewer 2 Report

My concerns have been addressed satisfactorily.
